# Characterization of a spontaneous microphthalmia-like mutant mouse with isolated ocular defects

**Jianying Wang**[1,2☉], **Fei Gao**[3☉], **Yuqiang Zheng**[3], **Mingqi Zhang**[3], **Yuzhu Zhou**[1], **Zhuoshi Wang**[3]*, **Jun Li**[1,2,3]*

**1** Jinzhou Medical University, Jinzhou, China, **2** He Eye Specialist Hospital, Shenyang, China, **3** He University, Shenyang, China

☉ These authors contributed equally to this work.
* dr.wangzs@foxmail.com (ZW); robin_lijun@sina.com (JL)

## Abstract

Microphthalmia is a significant eye defect owing to its profound effects on visual acuity. Microphthalmia accounts for 3.2%–11.2% of blind children. To date, there has been no cure for this disease. In this study, we aimed to identify microphthalmia-like mutant mouse and study its growth and development. In this study, we identified mutant mice exhibiting eye abnormalities using a forward genetics approach in a C57BL/6J cohort. To identify ocular characteristics of the mutant mouse, we conducted systematic evaluations including basic measurements (body length, body weight, and palpebral fissure width), optical coherence tomography (OCT), optomotor response (OMR), and hematoxylin-eosin (H&E) staining. At early developmental stages, there are notable differences in body length and weight between mutant and normal mice. Mutant mice displayed microphthalmia-like phenotypes, characterized by significantly reduced eyeball and lens sizes as well as decreased anterior chamber depth compared to wild-type controls. Visual impairment was evident in the mutant mice. Mutant mice exhibited rosette-like structures in the retina without impacting other organs of the body. Overall, these results support microphthalmia-like mutant mouse as a valuable tool for studying this congenital ocular malformation.

## Introduction

Microphthalmia is a severe ocular developmental disorder [1]. Microphthalmia is defined as a reduced size of the ocular globe in the orbit and is associated with a reduced corneal diameter (corneal horizontal diameter <9 mm in a newborn and < 10 mm in children >2 years) [2]. Microphthalmia is diagnosed as an axial length of <19 mm in a 1-year-old child or <21 mm in an adult using ultrasound B-scan and characterized by an axial eye length that is at least two standard deviations below the mean for age [2]. Microphthalmia is caused by perturbation of early eye organogenesis

**Data availability statement:** All relevant data are within the paper and its Supporting information files.

**Funding:** This research was funded by "The general funding program projects of the Joint Fund for 2024(2023-MSLH-079)" of the Department of Science and Technology of Liaoning Province and "Re-search Platform Construction Project (LJ232413610013)" of the Liaoning Education Department.

**Competing interests:** The authors have declared that no competing interests exist.

through genetic or environmental factors in the first trimester of pregnancy [3]. Although the majority of pathogenic mutations associated with non-syndromic microphthalmia occur sporadically through de novo mechanisms, and include missense, nonsense, frameshift, and splice-site variants, microphthalmia is inherited as autosomal-dominant, autosomal-recessive, or X-linked [2,4]. Vision impairment is the most severe case of microphthalmia. Microphthalmia is reported in up to 11% of blind children [3,5,6]. Currently, there is no cure for this disease. Clinical patients exhibit significant phenotypic heterogeneity and present varying severities of eye developmental disorders. Simple microphthalmia refers to a structurally normal, small eye and may be unilateral or bilateral [6]. Complex microphthalmia may also be associated with other ocular abnormalities such as cataract, anterior segment dysgenesis, retinal dysplasia or folds, or glaucoma, and 80% of patients exhibit systemic features [2]. The pathophysiological mechanisms underlying microphthalmia have not been fully elucidated, and hypotheses for disruptive mechanisms include deficient inductive signaling at the primitive neural tube level or a failure of the optic pit expansion and subsequent optic vesicle formation or secondary regression of established ocular structures during development [7].

Eye morphogenesis is a highly conserved development process among vertebrates [8,9]. Vertebrate eye development is regulated by highly conserved gene regulatory networks of eye region transcription factors (EFTFs) that control correct gene expression during early eye development, and mutations in the genes encoding EFTFs lead to ocular maldevelopment [10]. Ocular development initiates with the establishment of the eye field, a specialized cellular domain located in the anterior neural plate [9]. In the anterior neural plate, the eye-field is specified by the coordinated expression of the EFTFs [11]. Briefly, the ocular field subsequently undergoes bilateral division, with cellular extensions migrating laterally from the neural plate's midline to initiate evagination toward the surface ectoderm [3,11]. Concurrently, molecular signals originating from the optic vesicle induce thickening of a specialized pre-placodal domain within the surface ectoderm [3,12]. Reciprocal interactions between the emerging lens placode and optic vesicle drive synchronized invagination processes, culminating in lens pit formation [4]. This morphogenetic process establishes a bilaminar optic cup, with the outer layer differentiating into retinal pigmented epithelium (RPE) precursors and the inner layer giving rise to neural retina (NR) progenitors [13]. Progressive invagination of the lens pit ultimately generates the lens vesicle structure [4]. Following gestational day 47, retinal differentiation commences as the optic cup's outer layer matures into functional RPE, while the inner layer develops into organized NR tissue [4].

Investigating human ocular development poses morphological and molecular challenges, particularly regarding microphthalmia occurring during 4–8 weeks gestation, which disrupts normal eye formation [3]. Consequently, much of our understanding of microphthalmia stems primarily from animal mutant. Animal model play a crucial role in exploring pathogenic mechanisms underlying inherited human disease, and mice are the most common animal mutant for biomedical research [14]. 99% of the mouse genome is conserved compared with

humans, and eye development and mature ocular structure between humans and mice are similar [3]. Analysis of orthologous gene variants in animal model identified overlapping phenotypes for most factors, confirming functional conservation throughout vertebrate developmental processes [15]. Animal model facilitate further investigation of the mechanisms of microphthalmia, enabling the systematic organization of diverse genetic factors into coherent developmental networks [15].

While targeted mutagenesis has gained prominence for enabling direct modification of specific genes in mouse mutant generation, they are less ideally suited to testing the genetic basis of a specific phenotype of interest due to a lack of prior knowledge of the role of a gene of interest in a phenotype to be analyzed [16]. However, forward genetics approaches employed random mutagenesis in animal mutants followed by phenotypic screening to identify causative mutations, establishing many foundational microphthalmia mutants, including Mitf, first identified in the early mi mouse line [3]. In addition to spontaneous mutations found in the mouse Mitf gene, mutations have been found that were induced by irradiation and chemical mutagens, as exemplified by pathogenic variants in the mouse Mitf gene [17]. Forward genetics involves screening mice for mutant phenotypes without prior knowledge of their genetic basis and plays an important role in the generation of mutant mice [18]. A critical regulator, Pitx3, encoding the paired-like homeodomain transcription factor 3, was revealed through spontaneous mouse mutants (Pitx3$^{416insG}$ line) exhibiting closed eyelids with anophthalmia or microphthalmia [19,20].

In this study, our purpose was to identify the microphthalmia-like mutant mouse, which was discovered through natural screening and likely represents a spontaneous genetic mutant. The microphthalmia phenotype has been comprehensively characterized in mutant mouse, making it a tool for studying this severe eye disease.

## Materials and methods

### Generation of the mutant mice

Mutant mice were compared with age-matched C57BL/6J controls. Mice were housed under controlled environmental conditions at the He University's Experimental Animal Center, maintained in standardized polypropylene cages with corn-cob bedding, with ad libitum access to food and water under a 12-hour light/dark cycle. This study was carried out in strict accordance with the recommendations in the Guide for the Care and Use of Laboratory Animals of the National Institutes of Health. The study was approved by the Ethics Committee of Experimental Animal of He University (Shenyang, China) (protocol code: 2025031201 and date of approval:18 April,2025). (S1 Checklist).

A spontaneous microphthalmic mutant mouse was identified as the founder of the study. This founder was initially outcrossed with C57BL/6J mice to generate the F1 generation. All F1 mice displayed normal ocular morphology, indicating the absence of microphthalmia and suggesting that the mutant trait is not dominant. Phenotypically normal F1 heterozygous mice were intercrossed to produce the F2 generation. To confirm the inheritance patterns, F1 heterozygous mice were backcrossed with this founder, resulting in the BC1 generation. Selective intercrossing of microphthalmic mice from the F2 generation was conducted, and this inbreeding was maintained for 10 consecutive generations (F3–F10) to stabilize the phenotype until consistent microphthalmia expression was achieved in the same litter.

Statistical analysis of the F2 generation indicated that 45 out of 186 offspring displayed microphthalmia, resulting in a normal to mutant ratio of approximately 3:1 ($\chi^2$ test, P > 0.05). This observation aligns with Mendelian inheritance patterns of a single autosomal recessive locus. Furthermore, the backcross (BC1) generation revealed that half of the microphthalmic mice were present, providing additional evidence that the trait is governed by a recessive allele, as homozygosity for the mutant allele is necessary for phenotypic expression.

We systematically evaluated penetrance and expressivity across the F2–F10 generations. All homozygous mutant mice displayed the microphthalmia phenotype, and no phenotypically normal mice were identified among the genetically homozygous mutants. This finding indicates complete penetrance (100%) of the trait. To assess phenotypic

consistency, we quantified key ocular parameters, including equatorial diameter and lens diameter, in 30 homozygous mutant mice from the F5–F10 generations. No significant variation in the severity of microphthalmia was detected within the same litter or across generations. This observation demonstrates consistent expressivity of the phenotype, with no variability in severity (S2 Fig).

### Animal's basic data

Growth parameters (body length, weight and width the palpebral fissure) were systematically quantified at designated postnatal timepoints (P14, P28, P42, P56, P3M, P4M) in both control and mutant group, with statistical analyses rigorously performed. Ocular morphology in mice was documented through photographs. The width of the palpebral fissure, an anatomical parameter, was operationally defined as the linear distance spanning from the medial to lateral canthal junctions.

### Measurement and photograph of eyeball size

Equatorial diameter (from nasal to temporal) and lens diameter were measured following previously published protocols [21]. Normal and mutant mice were euthanized, and their eyes were enucleated, with extraocular connective tissues removed in phosphate-buffered saline (PBS). The eyeballs were positioned on quadrille paper for measurement, and the lenses were subsequently removed and placed on the paper. The sizes of the eyes and lenses were approximated using a ruler. Each group consisted of a minimum of six eyes for analysis.

### Anterior segment optical coherence tomography (AS-OCT)

The ocular anterior segment parameters were measured using anterior segment-specific OCT system (Carl Zeiss Meditec AG, Jena, Thuringia, Germany). Mice were anesthetized with an intraperitoneal injection of tiletamine hydrochloride and zolazepam hydrochloride (40 mg/kg; Zoletil 50, Vibrac, Carros, France). Pupillary dilation was performed with 0.5% tropicamide and 0.5% phenylephrine mixed eyedrop(Compound Tropicamide Eye Drops, Xingqi Eye Medicine Co, Ltd., Shenyang, China). To prevent corneal dehydration and reduce imaging interference, a thin layer of 0.3% hydroxypropyl methylcellulose gel was evenly applied to the corneal surface to maintain corneal moisture and transparency. This system equipped with a small animal-specific fixation platform and head positioning device was used. Light source wavelength of 840nm, axial resolution of 5μm, and lateral resolution of 15μm; the "anterior segment panoramic scanning mode" was selected, with a scanning range covering the cornea, anterior chamber, iris, and anterior surface of the lens. The scanning depth was set to 6 mm, single scan time to 0.5s, and each eye was scanned repeatedly 3 times to obtain stable images. OCT imaging provided detailed images of the anterior chamber. The anatomical anterior chamber depth (from the anterior surface of the lens to the posterior surface of the cornea) was quantified using the InSight-Animal OCT Segmentation Software (Phoenix Research Labs, Pleasanton, CA, USA).

### Optomotor response

The individual mice were positioned on a platform at the center of the OptoTrack visual detection system (XR-OT101, Xinruan, Shanghai, China). Four screens, monitored by a computer, displayed moving black-white vertical gratings that rotated continuously in a specific direction. Mice typically turned in the direction of the moving gratings under normal conditions. The Optomotor Response system quantified compensatory head movements when the direction of the moving gratings was reversed either clockwise or counterclockwise. Visual stimuli with maximum contrast (100%) were presented at four different spatial frequencies (0.05, 0.1, 0.2, and 0.3 cycles/°). Additionally, visual stimuli at the optimal spatial frequency (0.2 cycles/°) were presented at five different contrasts (100, 50, 25, 12.5, and 5%). Each stimulus at a specific spatial frequency or contrast level lasted 45 seconds and was presented in a randomized order and direction over five

trials. The number of head movements in response to stimuli moving in the same direction (T_Correct) and in the opposite direction (T_Incorrect) was recorded to calculate the optomotor response indices (T_Correct/T_Incorrect) at each spatial frequency and contrast level.

### Electroretinography (ERG)

ERG recordings were conducted under dark-adapted conditions. Mice were dark-adapted overnight. Before anesthesia, the pupils were dilated with 0.5% tropicamide and 0.5% phenylephrine mixed eyedrop (Compound Tropicamide Eye Drops, Xingqi Eye Medicine Co, Ltd., Shenyang, China). Mice were anesthetized following the protocol established in OCT procedure. After anesthesia, a ground electrode needle was placed in the tail, and a reference electrode needle placed in the face and applied hypromellose 2.5% (Goniovisc) on the surface of the cornea. Scotopic ERG was obtained at flash intensities of 1.5 cd·s/m² and 3.0 cd·s/m². B-wave amplitude was measured from the a-wave trough baseline to the peak of b-wave, and b-wave amplitude was measured from the onset of the stimulus to the a-wave trough.

### Hematoxylin and eosin staining (H&E)

Mice's eyes and organs at various developmental stages were harvested and fixed in Davidson's fixative solution for 24 hours or 4% paraformaldehyde fixative solution for 48 hours. The viscera, comprising the heart, liver, spleen, lung, and kidney, were included in the fixation process. Subsequently, the eye tissues and viscera underwent dehydration in ethanol gradients, clearing in xylene, and embedding in paraffin. Tissue sections of 5 μm thickness were obtained using a microtome (RM2245, Leica, Germany). For H&E staining, the sections were deparaffinized, hydrated in ethanol, and stained with hematoxylin and eosin. Each section was mounted with neutral balsam (WG10004160, Servicebio, China) and examined under a light microscope (ECLIPSE NI, Nikon, Japan).

### Statistical analysis

All quantitative analyses were executed through the implementation of GraphPad Prism 6. Representative data are presented as means±SEM. Intergroup comparisons were statistically evaluated using t-tests ($p < 0.05$ indicates statistical significance).

## Results

### Identification of mutant mice of microphthalmia-like

We developed a novel spontaneous microphthalmia-like mutant mouse. To visually exhibit the ocular phenotype of microphthalmia-like of mutant mouse, we performed ocular morphological observation. Characteristic manifestations of mutant mice included significantly reduced size of the ocular globe, narrowed palpebral fissures, diminished orbital volume compared to controls. As shown in Fig 1A, mutant mice maintained consistent microphthalmia-like phenotype in relevant development stages.

### The extra-ocular characterization of mutant mice of microphthalmia-like

The body length and weight of the control group and mutant group were measured at defined postnatal timepoints, and all representative values are expressed as the mean ± SEM (Tables 1 and 2). The body length of the mutant group was smaller than that of the control group from 28 days to 56 days. The weights of the mutant group and control group have no significant difference, except for the 28th day and 42th day (Fig 1B, C).

To exhibit ocular statistical differences between the control group and mutant group, the width of the palpebral fissure of the control group and mutant group was measured, and representative data are presented as mean ± SEM (Table 3). The

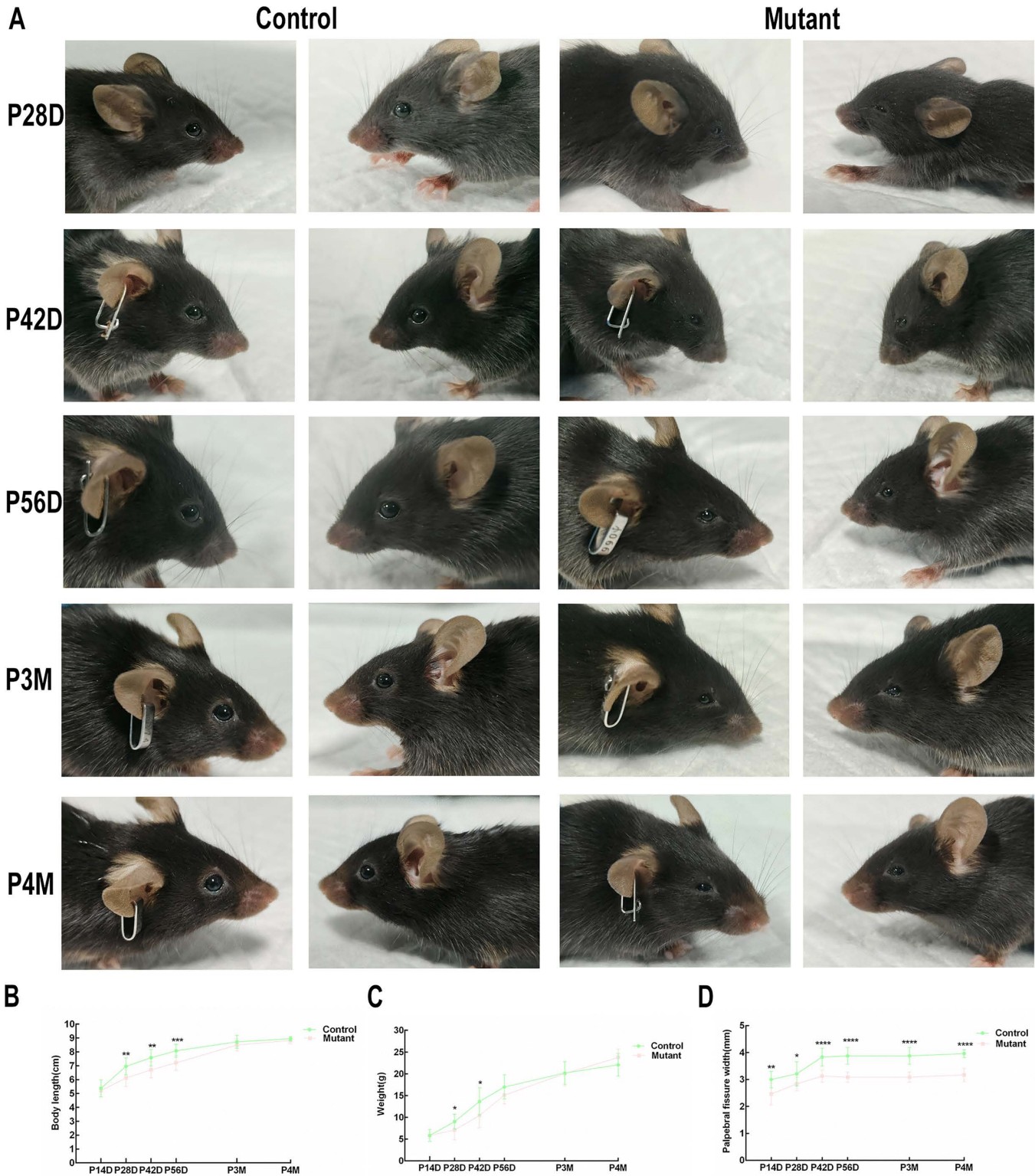

**Fig 1. The mouse mutant displays the ocular phenotype of microphthalmia-like. (A)** External morphology of the eyes of mutant mice and normal mice was photographed at P14, P28, P42, P56, P3M, or P4M. At each time point, bilateral ocular imaging was simultaneously performed in each group. The images are representative of results obtained in all 3 mice per group. Mice were positioned the same distance from camera for each image. **(B)** The

body length of the mutant group(n = 12) and the control group(n = 12). **(C)** The weight of the mutant group(n = 12) and control group(n = 12). **(D)** The width of the palpebral fissure of the mutant group (n = 12) and control group (n = 12). The width of the palpebral fissure of the mutant group was significantly smaller than that of the control group at each stage. Multiple t tests were performed to analyze the significance. Data are presented as the means ± SEM at three times. *p < 0.05, **p < 0.01, *** p < 0.001, **** p < 0.0001.

**Table 1. Body length (cm).**

| Day | Control | Mutant | P value | Control vs. Mutant |
|-----|---------|--------|---------|--------------------|
|     | (N = 12) | (N = 12) |         | P value |
| 14 | 5.38 ± 0.61 | 5.21 ± 0.26 | 0.391151 | n.s. |
| 28* | 6.96 ± 0.62 | 6.13 ± 0.61 | 0.00307448 | <0.01 |
| 42* | 7.58 ± 0.60 | 6.71 ± 0.58 | 0.00146035 | <0.01 |
| 56* | 8.08 ± 0.47 | 7.21 ± 0.54 | 0.000343297 | <0.001 |
| 90 | 8.73 ± 0.47 | 8.50 ± 0.43 | 0.23595 | n.s. |
| 120 | 8.95 ± 0.15 | 8.83 ± 0.25 | 0.173932 | n.s. |

* Statistical significance. Data are presented as mean ± standard error of the mean. n.s. represents not significant.

**Table 2. Weight(g).**

| Day | Control | Mutant | P value | Control vs. Mutant |
|-----|---------|--------|---------|--------------------|
|     | (N = 12) | (N = 12) |         | P value |
| 14 | 5.83 ± 1.35 | 5.84 ± 0.75 | 0.985221 | n.s. |
| 28* | 9.01 ± 1.72 | 7.08 ± 2.28 | 0.0286383 | <0.05 |
| 42* | 13.65 ± 3.17 | 10.43 ± 2.83 | 0.0155514 | <0.05 |
| 56 | 16.97 ± 2.78 | 15.11 ± 1.99 | 0.0727951 | n.s. |
| 90 | 20.18 ± 2.61 | 20.03 ± 2.81 | 0.897089 | n.s. |
| 120 | 22.06 ± 2.61 | 23.77 ± 1.90 | 0.0857102 | n.s. |

**\*** Statistical significance. n.s. represents not significant.

**Table 3. Palpebral fissure width(mm).**

| Day | Control | Mutant | P value | Control vs. Mutant |
|-----|---------|--------|---------|--------------------|
|     | (N = 12) | (N = 12) |         | P value |
| 14* | 3.00 ± 0.30 | 2.41 ± 0.38 | 0.00106265 | <0.01 |
| 28* | 3.21 ± 0.45 | 2.82 ± 0.25 | 0.0189971 | <0.05 |
| 42* | 3.83 ± 0.33 | 3.09 ± 0.20 | < 0.0001 | <0.0001 |
| 56* | 3.88 ± 0.31 | 3.05 ± 0.15 | < 0.0001 | <0.0001 |
| 90* | 3.88 ± 0.31 | 3.05 ± 0.15 | < 0.0001 | <0.0001 |
| 120* | 3.96 ± 0.14 | 3.14 ± 0.23 | < 0.0001 | <0.0001 |

* Statistical significance.

width of the palpebral fissure of the mutant group was smaller than that in the control group at each stage. The width of the palpebral fissure of the mutant group and control group exhibited progressive expansion until P42, followed by stabilization of ocular parameters thereafter (Fig 1D).

## The ocular characterization of mutant mice of microphthalmia-like

To further identify the ocular phenotype of mutant mice with microphthalmia-like, eyeballs and lenses were enucleated and photographed from the control group and mutant group, and then was estimated by ruler and quadrille paper (Fig 2A). The size of the eyeballs and lenses of the mutant group is significantly smaller than that of the control group (Fig 2B).

## Vision function of mutant mice

To assess the vision function of mice, we utilized the OptoTrack visual detection system, which captures optomotor responses through video tracking algorithms. Optomotor response (OMR) is quantified as the ratio of response times between head movements in correct and incorrect directions. OMR values were measured under a wide range of velocity thresholds at defined spatial frequency or contrast levels [22]. At the same spatial frequency, the overall OMR values in the control group appeared relatively higher level than those of the mutant group. In the control group, the overall OMR values at a spatial frequency of 0.1 cycles/° show a peak and were higher than 1.8 at 0.1 cycles/° (presented in red) at the different spatial frequencies (Fig 3A). The OMR value in the mutant group at a spatial frequency of 0.05 cycles/°reached 1.46 (Fig 3B).As for the contrasts, the overall OMR values started with the highest level at 100% contrast, followed by a decline afterward in both groups (Fig 4A). However, the overall OMR values in the mutant group initially peaked and then gradually decreased at defined spatial frequency or contrast levels.

Friedrich Kretschmer et al. established that constraining stimulus correlated head movements within the 2–14°/s velocity range as the ideal velocity threshold criteria, which is likely to effectively minimize head movements unrelated to visual stimuli [22,23]. We analyzed optomotor response within this velocity threshold range (the color of the bar is presented in magenta and green). To ensure uniform evaluation of experimental data, we standardized the results. The response time in correct or incorrect directions at each velocity interval (1°/s per interval) was normalized to the maximum response time (represented as 1) for each spatial frequency or contrast. Visual stimuli–driven responses at a spatial frequency of 0.05 cycles/° in the mutant group were more frequent and lasted longer (blue bars) in the stimulus direction (positive value, light green window) compared to the incorrect direction (negative value, light magenta window). For the contrasts, the OMR value in the mutant group appeared maximal at 100% contrast and was equal to 1.26 (Fig 4B). Visual stimuli–driven responses in the mutant group exhibited a relatively smooth pattern of change in spatial frequency and contrast level (Fig 4C, D).

## The anterior chamber depth of mutant mice of microphthalmia-like

Anterior segment dysgenesis is one of the other ocular abnormalities associated with complex microphthalmia. For quantitative assessment of anterior chamber depth, anterior segment OCT scans were systematically employed for mice ocular imaging. The clear anterior segment images were acquired to evaluate anterior chamber depth at the P4M observation endpoint. Anterior chamber depth in the mutant group was shorter than that in the control group (Fig 5A). Furthermore, we measure the corneal thickness. Through measurement and statistical analysis of corneal thickness, the results showed that the mutant group did not exhibit significantly thinner corneas compared to the control group (Fig 5B).

To further assess the specific visual function of the mutant retina, we performed electroretinography (ERG) analysis under dark adaptation conditions using flash intensities of 1.5 cd·s/m² and 3.0 cd·s/m². The mutant mice exhibited abnormal ERG waveforms (Fig 6A). The a-wave and b-wave amplitudes in the mutant groups were significantly reduced compared to the control group at flash intensities of 1.5 cd·s/m² and 3.0 cd·s/m², respectively (Fig 6B). The results indicated that the mutant group exhibited significantly compromised visual function compared to the control group. However, the underlying mechanism for the observed visual impairment in mutant mice remains incompletely understood. This functional deficit may originate from intrinsic retinal pathology, or alternatively, result from the physical obstruction of light entry caused by pupillary atresia in mutant mice (Fig 6C).

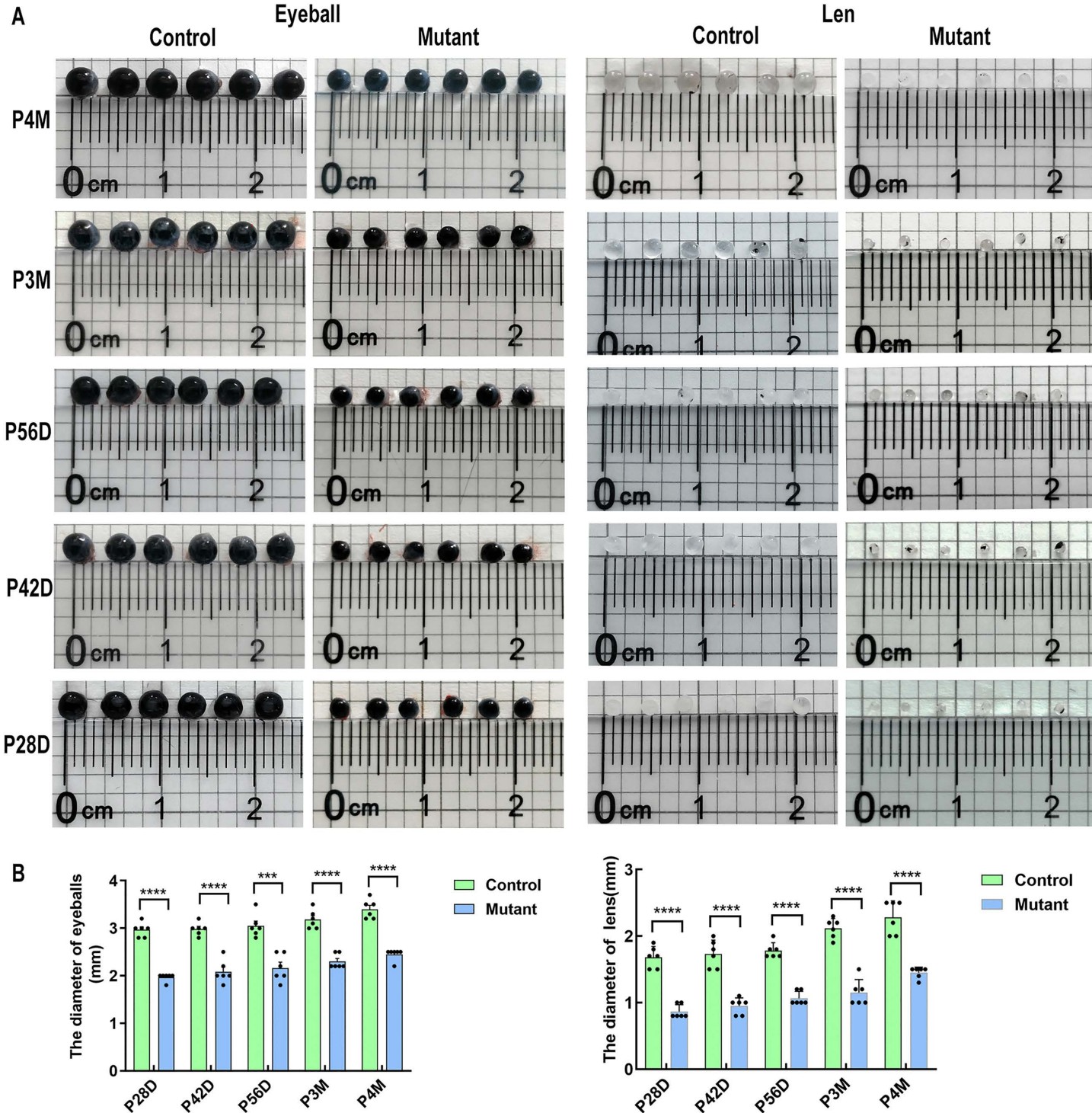

**Fig 2. The size of the eyeballs and lenses of the mutant group and the control group. (A)** Six eyeballs and six lenses of the mutant group and control group at designated postnatal timepoints (P28, P42, P56, P3M, P4M) are presented on quadrille paper. **(B)** The size of the eyeballs and lenses were quantified and multiple t tests were used for statistical analysis. The eyeballs and lens of the mutant group(n = 6) are significantly smaller than those of the control group (n = 6). **** $p < 0.0001$.

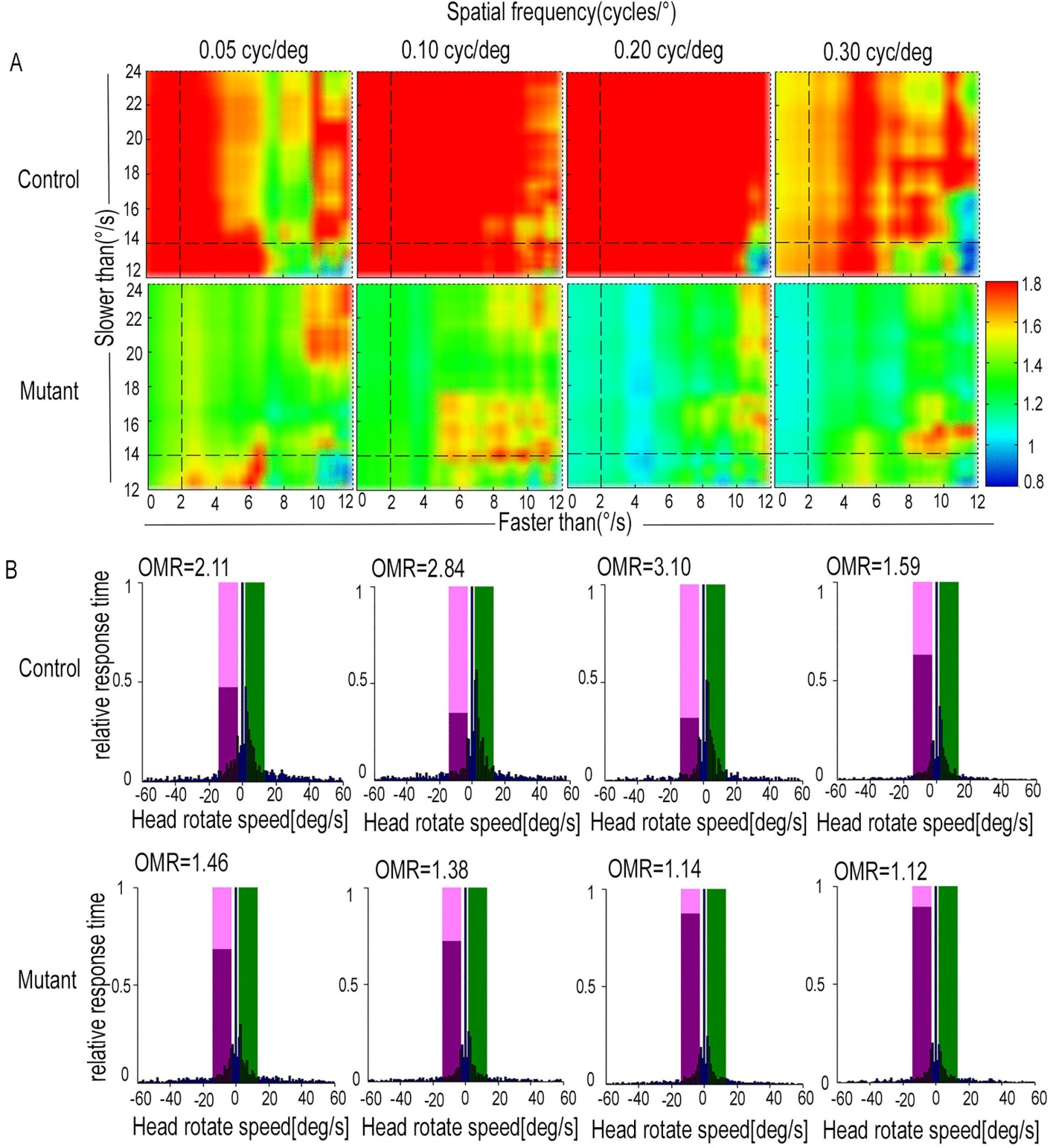

**Fig 3. Optomotor response behavior test of the mutant group and control group at defined spatial frequency. (A and B)** Vision function was detected at defined spatial frequency or contrast levels in the OptoTrack visual detection system. **(A)** Representative heat maps show the OMR values at defined spatial frequencies for both the mutant and control groups at P3M. **(B)** The overall OMR values in the mutant group were significantly lower than those of the control group at defined spatial frequencies.

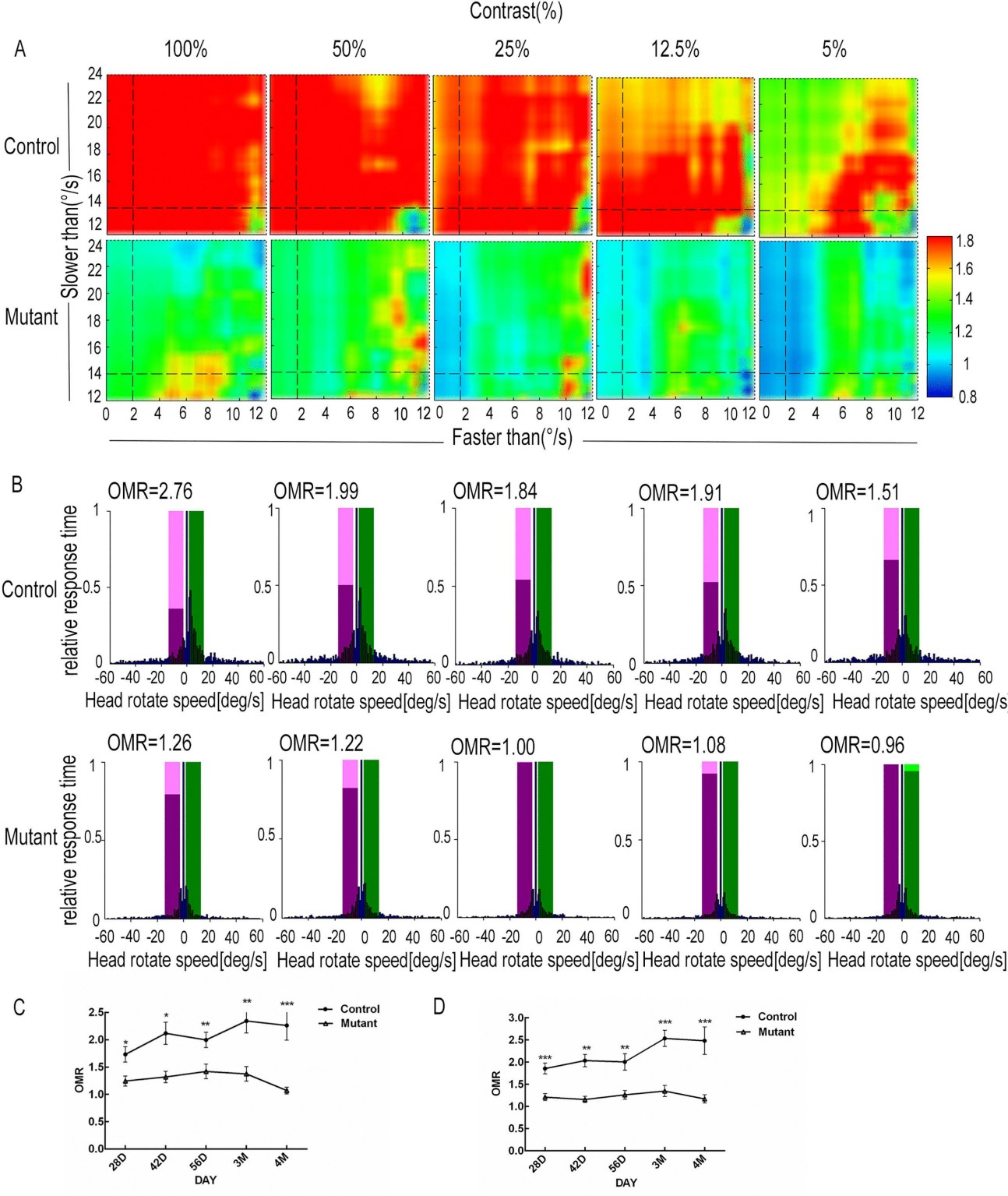

**Fig 4. Optomotor response behavior test of the mutant group and control group at defined contrast. (A)** Representative heat maps show the OMR values at defined contrast at P3M. **(B)** Representative histograms of visual stimuli–driven response of the mutant group at defined contrasts under the optimal velocity conditions (2–14°/s) at P3M. The response time in correct (positive value, light green window) or incorrect direction (negative

value, light magenta window) within the defined velocity interval was normalized against the maximum response time (represented as 1) for each spatial frequency or contrast and depicted as a blue bar. In cases where the response time in the correct direction exceeds that in the incorrect direction, the OMR values for each spatial frequency or contrast are illustrated as a magenta bar normalized to the green bar and also the green and light green bars exhibit equivalent vertical dimensions, resulting in the light green bars are covered by green bars. **(C and D)** The mean OMR values (mean ± SEM) of the mutant group (n = 12) and control group (n = 15) were plotted as a line chart at a spatial frequency of 0.1 cycles/°(C)or 100% contrast **(D)**. The mean OMR values of the mutant group and control group have a significant difference at spatial frequency of 0.1 cycles/°or 100% contrast. Multiple t tests were performed in representative data, *p < 0.05, **p < 0.01, *** p < 0.001. All experiments were independently performed at least three times to ensure repeatable results.

## Histopathological change of mutant mice of microphthalmia-like

To verify whether the mutant mice exhibit systemic health differences compared to wild-type controls, we conducted a comprehensive baseline health assessment. We found that mutant mice and wild-type mice exhibit a well-groomed, sleek coat without lesions and normal exploratory behavior. Vital parameters, such as heart rate and body temperature, remain within established normal ranges. The animal exhibits a healthy appetite and consistent water consumption and stable body weight. Analysis of this baseline assessment demonstrated no statistically significant variation between the groups.

To analyze histological abnormalities, we utilized eye slices and tissue slices from paraffin sectioning to conduct pathological observations via H&E staining. Histopathological evaluation using H&E staining showed the retina of the mutant group and control group at P28, P42, P56, P3M and P4M. Our findings reveal a decrease in size in the eyeballs, retinal folds, and rosette-like structures in the mutant group compared to the control group (Fig 7A).

Histological analysis by HE staining demonstrated a significant thickening of the retinal pigment epithelium (RPE) in mutant mice compared with controls (Fig 7B). Furthermore, the high density of retinal rosettes in certain regions of the mutant mouse retina makes it challenging to accurately measure the thickness of the outer nuclear layer (ONL). Therefore, we quantified the number of retinal rosettes at multiple time points. A pronounced retinal fold is observed near the optic disc, whereas the peripheral retina appears normal. The results indicated that the outer nuclear layer of the mutant group exhibited varying degrees of severe pathological alterations compared to the control group at various time points (Fig 7C). However, viscera tissues of the mutant group showed no pathological changes (Fig 8A). Based on these results, we speculate that the pathological alterations in this mouse model were confined to the eyes and did not impact other regions.

## Discussion

This investigation identified and characterized a mutant mouse of microphthalmia-like and systematically evaluated its developmental impacts on ocular and other organs. Phenotypic analysis confirmed relative stability of the ocular abnormality across all mutant progeny, with ocular external morphology demonstrating stable microphthalmia features throughout postnatal development stages compared to controls. It remains unclear whether the microphthalmia phenotype is evident in ocular development stages of embryo. Quantitative analysis revealed significant disparities in body length between the mutant and control group from postnatal day 28–56 (p < 0.05), while body weight measurements showed significant differences at timepoints (P28 and P42). Therefore, the discrepancies between the mutant group and control group were only observed at the early stage of development. Due to the impaired vision function of mutant mice, the possible hypothesis is that after birth, visual impairment in both mother and offspring mice leads to impaired nursing efficiency during the lactation period, resulting in nutritional deprivation; after weaning, offspring mice compensate for growth through active feed intake. It still requires further long-term observation to verify this hypothesis.

In addition, comparative analysis of the experimental data revealed significantly impaired vision function of the mutant group compared to the control group by behavioral oculomotor test. Also, to further characterize retinal function, we performed electroretinography (ERG), which revealed significant visual function impairment in the mutant mice. However, due to the presence of pupil atresia in some mutant mice, the specific cause of visual function impairment remains

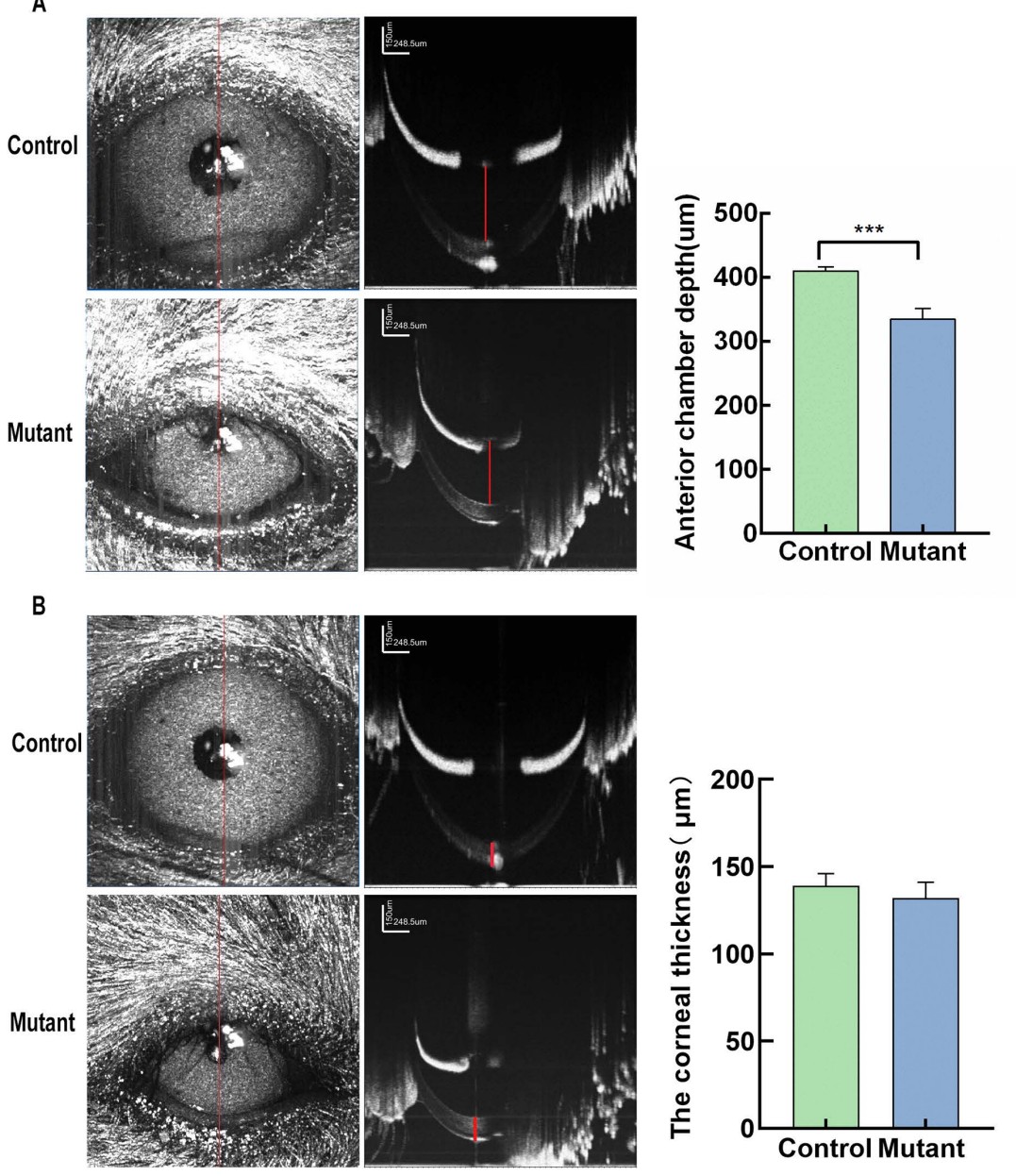

**Fig 5. Measurement of anterior chamber depth and corneal thickness in the mutant group and control group. (A)** Representative OCT image of the anterior segment of the mutant group(n=8) and control group(n=8). Anterior chamber depth was presented as a red line. Scale bar: 150μm. Anterior chamber depth in the mutant group was significantly shorter than that in the control group. ***p<0.001 **(B)** Representative OCT image of the corneal thickness of the mutant group(n=8) and control group(n=8). The corneal thickness is marked with a red line. Scale bar: 150μm.

unclear. Cheryl Y. et al. demonstrated that the localized administration, START (0.9% sodium chloride, 1% Tween 80, 1% powdered ataluren, 1% carboxymethylcellulose), effectively ameliorated ocular abnormalities, including corneal, lens, and retinal defects in small eye (Pax6Sey+/–) model of aniridia [24]. Their research further demonstrated the therapeutic potential of this treatment through the successful restoration of both electrical and behavioral responses of the retina [24].

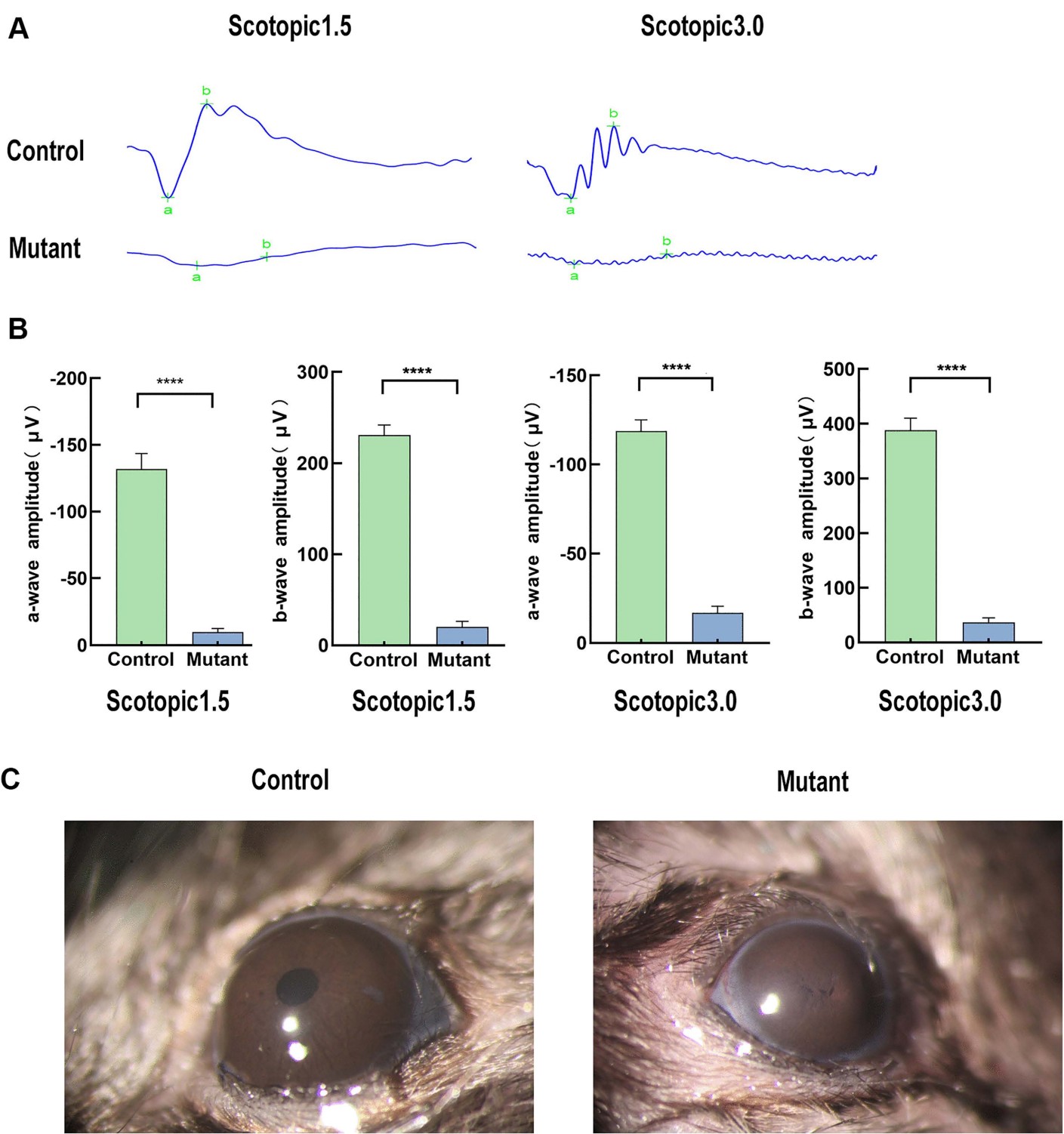

**Fig 6. Electroretinography (ERG) recording analysis of the mutant group and control group. (A)** A thorough evaluation of visual function using ERG was conducted in mice. Representative ERG waveforms from the mutant group (n = 8) and the control group (n = 8) are presented at P4M. **(B)** Quantitative analysis of a-wave and b-wave amplitudes was performed under dark-adapted conditions at flash intensities of 1.5 cd·s/m² and 3.0 cd·s/m², respectively. **** p < 0.0001 **(C)** Representative morphology of the eyes in the mutant group and the control group. Pupil atresia was noted in the mutant mice.

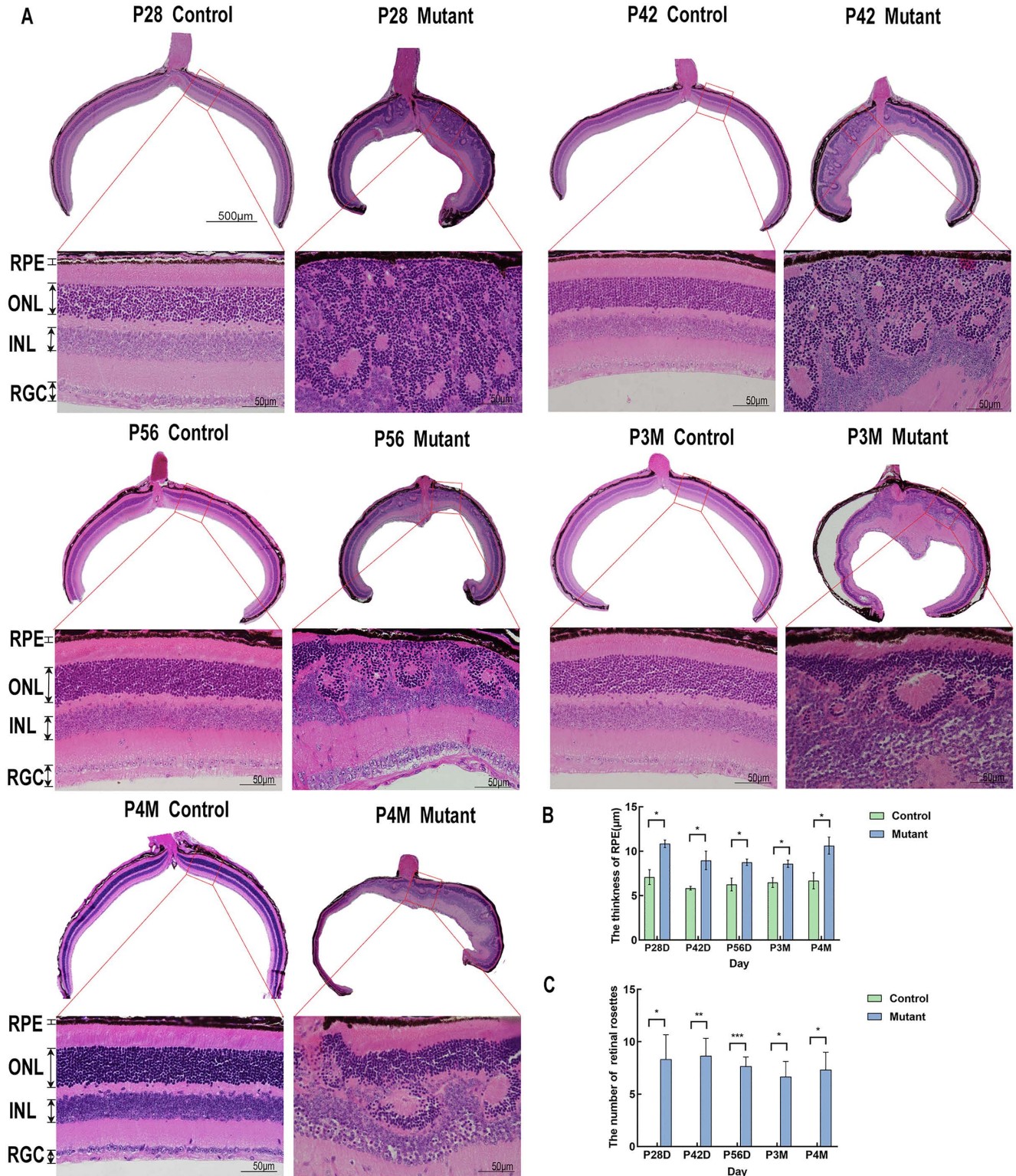

**Fig 7. H&E staining of eyeballs and of the mutant group and control group. (A)** Hematoxylin-eosin staining was performed on eye sections from the control and mutant groups at P28, P42, P56, P3M and P4M. A vertical section of the globe was taken through the optic disc plane. The red rectangle highlights areas that are depicted at higher magnification. Eyeballs in the mutant group exhibited the smaller size of the eyeballs and rosette-like

structures in the retina compared to the control group. The retinal layer identifications: retinal pigmented epithelium (RPE), outer nuclear layer (ONL), inner nuclear layer (INL), and retinal ganglion cell (RGC). Representative histological data of the retina of mutant mice(n = 3) and normal mice(n = 3). Scale bar: 50μm **(B)** Measurement and statistics of the thickness of RPE of mutant mice(n = 3) and normal mice(n = 3) at P28, P42, P56, P3M and P4M. **(C)** Count and statistical analysis the retinal rosettes in the vertical section of mutant mice(n = 3) at P28, P42, P56, P3M and P4M.

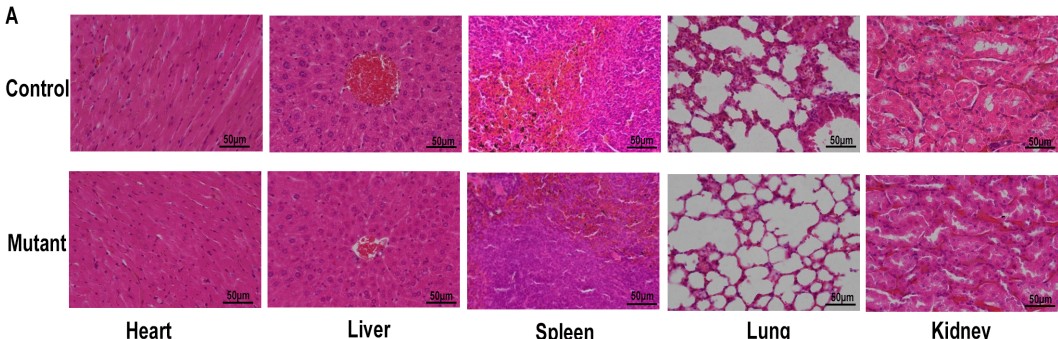

**Fig 8. H&E staining of viscera tissues of the mutant group and control group. (A)** HE staining was performed on tissues sections from the control and mutant groups at P4M. Representative image of viscera tissues of mutant mice(n = 3) and normal mice(n = 3). Scale bar: 50μm.

The study indicated that postnatal ocular interventions offer valuable insights for addressing microphthalmia in clinical settings. Moreover, the mutant group demonstrated significant ocular hypoplasia compared to control groups, characterized by reduced size of eyeballs, lenticular volume reduction, diminished anterior chamber depth, and pathognomonic retinal rosette formation. Notably, this investigation implemented an innovative measurement protocol for anterior chamber quantification, superseding conventional slit-lamp methodologies. The newly established methodology facilitated high-precision quantification of anterior-chamber depth.

Histopathological evaluation revealed organ-specific pathology restricted to ocular tissues, with preserved structural integrity observed in systemic organs (liver, cardiac muscle, pulmonary parenchyma, splenic architecture, kidney) in the mutant group. These findings indicate that gene mutation of mutant mice has only an impact on eye development abnormalities. Kim et al indicated that PXDN knockout mice exhibited completely or almost closed eyelids with small eyes, while demonstrating no significant abnormalities in other organ systems [25]. But their studies primarily focused on external morphological assessments in other organ systems, whereas our protocol incorporated systematic histopathological analysis using H&E staining to evaluate both microanatomical architecture of ocular and systemic tissues. Moreover, we observed the formation of retinal rosettes in mutant mice. Such rosettes have been extensively documented in various mouse models of inherited retinal diseases. EMC3 deficiency during retinal development induces the formation of retinal rosettes, a typical pathological structure observed in retinal disorders [26]. Retinal folds exhibit structural similarities to those found in syndromic oculo-skeletal dysplasia and present as invaginations and rosettes within the outer nuclear layer, which can be detected using spectral-domain optical coherence tomography (sdOCT) as well as in conventional H&E-stained sections from dogs affected by either acquired or inherited forms of the condition [27]. In the rd8 mouse model, CRB1—expressed in Müller glia and localized to the subapical region near the outer limiting membrane—is mutated, leading to disrupted retinal lamination and the formation of folds or pseudorosettes between the outer and inner nuclear layers [28]. MTHFR deficiency exacerbates the retinal pathology in Mthfr + / − rd8/rd8 mice—a model associated with Leber congenital amaurosis and retinitis pigmentosa—manifesting as focal dysplasia featuring profound disorganization with exaggerated rosettes amidst otherwise preserved retinal architecture [29]. The retinal rosettes and dysplastic features observed in our mutant exhibit significant pathological similarities to those in these models, indicating a possible

involvement of conserved pathways that regulate retinal lamination and cell polarity. In the Nrl knockout mouse, loss of neural retina leucine zipper (*Nrl*) function disrupts photoreceptor fate determination, causing rod precursors to undergo ectopic differentiation into cone-like cells and leading to the characteristic formation of retinal rosettes [30]. Mutations in other genes such as Dll1, Abca4, Nr2e3and PALS1 result in the formation of retinal rosettes [31–34], which is a phenotypic outcome of disrupted retinal developmental programming. The presence of retinal rosettes may indicate disturbances in fundamental developmental pathways.

The mutant mice exhibited a significantly thickened RPE layer compared with the control group. However, given the multifaceted physiological roles of RPE cells, their dysfunction and degeneration are strongly implicated in the pathogenesis of multiple retinal degenerative disorders, including retinitis pigmentosa, Best disease, age-related macular degeneration, and Stargardt disease [35]. Best's disease (BD), an autosomal dominant macular degeneration, is characterized by the accumulation of lipofuscin-like material within and beneath the retinal pigment epithelium (RPE), driving progressive central vision loss [36]. In AMD, RPE dysfunction drives the progressive accumulation of basal deposits (drusen), leading to hypoxia, choroidal neovascularization, and eventual degeneration of the RPE and photoreceptor cells [37]. In this study, the mutant mice also exhibited abnormalities in the RPE. The observed RPE thickening resembles the structural changes seen in related retinal diseases, further demonstrating that the mutant mice have defects in eye development which lead to impaired visual function.

While targeted knockout mutations resulting in loss-of-function provide crucial mechanistic insights, spontaneous and induced genetic variations frequently generate unexpected phenotypic manifestations, and these serendipitous outcomes often reveal novel biochemical pathways and functional relationships, particularly when multiple allelic variants coexist within a single genetic locus [38]. The elucidation of the discovery of the microphthalmia locus and its gene is a case in point, and despite nearly seven decades of research on Mitf, researchers maintain that definitive mechanistic insights remain elusive in this field [39]. Spontaneous mutations, as naturally occurring events, produce a full and unbiased array of mutation types – single nucleotide variants (SNVs), small insertions or deletions [40]. Consequently, spontaneous mutations can lead to a variety of effects on protein function, more accurately reflecting the mutations associated with human genetic diseases than the genetically engineered knockout mutations targeting the same gene [40]. While targeted knockout mutations predominantly induce complete gene inactivation, resulting in a null phenotype, hypomorphic (reduced function) phenotypes yielded by spontaneously occurring murine mutants can be significantly different than the null phenotype, thus revealing critical insights into gene function during developmental and pathological processes [16]. Spontaneous mutant mice result in phenotypes that closely resemble human pathological conditions, making them particularly valuable for studying disease mechanisms. Spontaneously occurring murine mutants of retinal degenerative diseases (RDD's) display pathophysiological hallmarks of similar human clinical manifestations, and a representative example is the Pde6b[rd1] strain, which harbors a nonsense mutation in the Pde6b gene that codes for the β-subunit of cGMP phosphodiesterase (PDE) [41]. This mutant exhibits striking phenotypic congruence with autosomal recessive retinitis pigmentosa patients, which is a result of cGMP-PDE gene mutations (OMIM180072) [41].

The deficiency of this study lies in the fact that the pathogenic gene of mutant mice has not yet been identified. Whole-genome sequencing (WGS) enhances variant detection capability across coding and noncoding regions of known and novel microphthalmia genes [7]; thus, we can employ WGS to screen candidate genes underlying microphthalmia. However, validating the extensive pool of candidate genes individually proves challenging. Therefore, we can briefly speculate on candidate genes based on phenotypic similarity to known microphthalmia mutations. Foxe3 mutants exhibit phenotypic parallels with Pitx3 mutants, including microphthalmia and aphakia, reflecting the phenotypic similarity of patients with pathogenic mutations: PITX3-associated anterior segment dysgenesis 2 (OMIM 610256) and FOXE3-associated cataract 11 (OMIM 610623), both characterized by microphthalmia, cataract, anterior segment disorders, and sclerocornea, suggesting conserved molecular regulatory mechanisms [7,42]. Mutations in these candidate genes typically manifest microphthalmia phenotypes, frequently co-occurring with other ocular abnormalities.

In this report, we developed and identified a novel mouse mutant exhibiting microphthalmia-like ocular anomalies. The spontaneous microphthalmia mutant model established in this study provides a reliable platform for investigating the pathological mechanisms of congenital ocular developmental defects. This model recapitulates key features of human microphthalmia (e.g., reduced ocular biometrics, retinal dysplasia, and visual function deficits) and avoids the potential biases of genetically engineered models, offering a unique tool to explore naturally occurring genetic perturbations in ocular development.

Identification of causal microphthalmia genes will provide insight into pathogenesis as well as potential therapeutic targets.

## Conclusions

This study identified a microphthalmia-like mutant mouse through phenotypic screening. The mutant represents a useful tool for investigating severe ocular malformations. Mutant mouse exhibited microphthalmia-like features. There is statistically significant disparities in body length and weight between the mutant and control group in the early stage of development. Functional assessments revealed significant visual impairment in the mutant group. Ocular malformations of the mutant mouse, including reduced size of eyeballs and lens, shallow anterior chamber depth, and rosette-like structures in the retina, were observed without impacting other organs of the body. The mutant mice are highly valuable for studying disease progression and the etiology of this ocular defect.

## Supporting information

**S1 Checklist. Author Checklist-Full.**
(PDF)

**S2 Fig. Schematic diagram of the breeding process of mutant mice.**
(TIF)

**S3 Table. Original date.**
(XLSX)

## Author contributions

**Conceptualization:** Fei Gao.

**Formal analysis:** Fei Gao, Mingqi Zhang.

**Investigation:** Yuqiang Zheng, Yuzhu Zhou.

**Methodology:** Yuqiang Zheng.

**Resources:** Zhuoshi Wang.

**Software:** Yuzhu Zhou.

**Supervision:** Mingqi Zhang, Zhuoshi Wang, Jun Li.

**Validation:** Jianying Wang.

**Writing – original draft:** Jianying Wang.

**Writing – review & editing:** Zhuoshi Wang, Jun Li.

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
