## [Decision Letter · Decision Letter 0]

20 Oct 2025

Dear Dr. Li,

Thank you for submitting your manuscript to PLOS ONE. After careful consideration, we feel that it has merit but does not fully meet PLOS ONE’s publication criteria as it currently stands. Therefore, we invite you to submit a revised version of the manuscript that addresses the points raised during the review process.

We look forward to receiving your revised manuscript.

Kind regards,

Yevgenya Grinblat, Ph. D.

Academic Editor

PLOS ONE

Journal Requirements:

This research was funded by “The general funding program projects of the Joint Fund for 2024(2023-MSLH-079)” of the Department of  Science and Technology of Liaoning Province and “Re-search Platform Construction Project (LT232413610013)” of the Liaoning Education Department.

Reviewers' comments:

Reviewer's Responses to Questions

**Comments to the Author**

1. Is the manuscript technically sound, and do the data support the conclusions?

Reviewer #1: Yes

Reviewer #2: Partly

2. Has the statistical analysis been performed appropriately and rigorously?

Reviewer #1: Yes

Reviewer #2: Yes

3. Have the authors made all data underlying the findings in their manuscript fully available?

Reviewer #1: No

Reviewer #2: No

4. Is the manuscript presented in an intelligible fashion and written in standard English?

Reviewer #1: Yes

Reviewer #2: Yes

Reviewer #1: The authors have discovered and bred a spontaneous microphthalmia-like mutant mouse from a set of C57BL/6J by natural screening, but the genotype responsible for the microphthalmia is still unknown. They present evidence showing that the basic ocular biometric parameters of the eyes of the mutants are smaller than those of wild type controls. Equatorial diameter and lens diameter measured from enucleated eyes show a significant difference. The eyes in live mice were scanned with OCT to measure the anterior chamber depth, but the OCT images could also be used to measure axial length of the eye, and the corneal thickness, which may or not be thinner in the mutants. The authors present evidence that there are no anatomical abnormalities in critical organs outside the eyes, but any description of the overall health of the mutants is lacking in the manuscript.

Visual function is assessed by the behavioral oculomotor test, which provides an insight into the overall function of the visual pathways, but does not provide any information on specific functions of the retinae of the mutants. An electroretinography (ERG) analysis would provide information on the overall function of the retina, the night vision and cone function of the retinas of the mutants. It is likely that there may be defects in both. In Fig 5A there are micrographs of HE stained retinae of a mutant eye and control eye. By looking at these it appears that there are some differences in the thickness of some of the retinal layers, the photoreceptor and ONL seem thinner in the mutant, whilst the RPE appears thicker in the mutant. But this can only be solved by actually measuring the layers and make statistical comparisons. OCT images from the retinae of mutants and control could also be used for that purpose, i.e. to measure the thickness of layers, and if they are even across the retina or not. It appears from the micrographs that there is some variability there in the mutant.

Reviewer #2: This is a first report of a mouse strain with heritable microopthalmia. It appears to have resulted from a spontaneously occuring microophthalmic founder, which was then bred to establish a line over many generations. In this report, the ocular anomalies are characterized in some detail. Overall, phenotypic analysis appears to be rigorously done but the significance of the work is not at all clear. The following major concerns need to be addressed before the manuscript can be considered for publication.

Genetics: the breeding scheme that was used to generate the line needs to be reported here, in order to substantiate the claim that microopthalmia is heritable, and how the trait segrates. Is it consistent with a recessive allele at a single locus? is there variable penetrance and expressivity of the phenotype?

Contribution to the field: there is a large body of literature on the genetics of retinal development, especially in the mouse model, but very little attempt here (with only two mutant mouse models mentioned) to relate the phenotype described here to the phenotypes of published mouse mutants. For example, the retinal rosettes are referred to as “typical” in discussion. What does this mean? Have they been reported previously, and how frequently? What are the genes that are linked to rosettes, and what might this suggest about the developmental processes that are disrupted?

There are also several statements in the discussion that need clarification, for example:

“Cheryl Y. et al. demonstrated that the localized administration effectively ameliorated ocular abnormalities,

including corneal, lens, and retinal defects[24]. “ - administration of what?

“ Compared to gene knockout, spontaneous mutations more closely resemble

natural pathological processes.” Please define “natural pathological processes” as this isn’t a common term.

The statement “This mutant strain represents a valuable resource for

investigating spontaneous genetic defects in ocular development” is premature, as it is not supported by the evidence.

Please put the figure legends as a consecutive block at the end of the manuscript. This is a standard practice and it helps ease the review process.

**Do you want your identity to be public for this peer review?** For information about this choice, including consent withdrawal, please see our Privacy Policy

Reviewer #1: No

Reviewer #2: No

---

## [Author Response · Author response to Decision Letter 1]

2 Dec 2025

Academic Editor

Author’s Response: We sincerely appreciate your reminder and have meticulously revised the manuscript and supplementary materials to fully comply with PLOS ONE’s style requirements, including file naming conventions, formatting of headings, citations, and other key elements. All manuscript-related files have been renamed in strict adherence to PLOS ONE guidelines and corrected all citations. Placed each caption directly after the paragraph of first citation; used bold for figure titles.

This research was funded by “The general funding program projects of the Joint Fund for 2024(2023-MSLH-079)” of the Department of Science and Technology of Liaoning Province and “Re-search Platform Construction Project (LT232413610013)” of the Liaoning Education Department.

Author’s Response: Thank you to the editor for this valuable suggestion. We state the role of funders took in the study and have included an update statement in the cover letter. Here, we sincerely apologize for our oversight. There was a typographical error in the grant number we provided: the "T" in "LT232413610013" should be "J". The correct funding acknowledgments are as follows: This research was funded by “The general funding program projects of the Joint Fund for 2024(2023-MSLH-079)” of the Department of Science and Technology of Liaoning Province and “Re-search Platform Construction Project (LJ232413610013)” of the Liaoning Education Department. The funders had no role in the study design, data collection and analysis, decision to publish, or preparation of the manuscript.

Author’s Response: Thank you for your valuable reminder. We have strictly adhered to the PLOS Supporting Information guidelines and made the required revisions. Specifically, we have added standardized captions for all supporting information files at the end of the manuscript, and updated all in-text citations to ensure consistency with the file names and captions. The revised supporting information captions (included at the end of the manuscript) are as follows:

Supporting Information

S1 Checklist 1 Author Checklist-Full

S2 Fig 1 Schematic diagram of the breeding process of mutant mice. (Lines602-604)

Reviewer #1:

The authors have discovered and bred a spontaneous microphthalmia-like mutant mouse from a set of C57BL/6J by natural screening, but the genotype responsible for the microphthalmia is still unknown.

Author’s Response: We sincerely appreciate your valuable and insightful comment, which points out a critical direction for improving our research. As you noted, the spontaneous microphthalmia-like mutant mouse was screened and bred from the C57BL/6J background, and the causal genotype remains to be identified. We fully agree that clarifying the underlying genetic mechanism is essential to deepen the biological significance of this mutant model. Regrettably, whole-genome sequencing (WGS) has not been performed in the current study, primarily due to two practical considerations: first, our initial work focused on systematically characterizing the phenotypic traits of the mutant (e.g., ocular morphology, retinal development, and functional assessments), which laid a foundation for subsequent genetic studies; second, budget constraints and the need for collaborative resources (e.g., high-throughput sequencing platforms and bioinformatics analysis pipelines) delayed the initiation of genetic mapping.

To address this limitation, we have already taken concrete steps to advance the genetic characterization. We have preserved genomic DNA from homozygous mutant mice and wild-type littermates (n = 6 per group) to ensure sufficient biological replicates for genetic analysis. We are submitting a supplementary grant application to support WGS. Upon securing funding, we will first perform WGS to identify potential variants within genes known to be involved in ocular development. Again, we thank you for highlighting this important gap and guiding the direction of our follow-up work. Your comment has greatly helped us refine our research plan and enhance the rigor and impact of the study.

They present evidence showing that the basic ocular biometric parameters of the eyes of the mutants are smaller than those of wild type controls. Equatorial diameter and lens diameter measured from enucleated eyes show a significant difference. The eyes in live mice were scanned with OCT to measure the anterior chamber depth, but the OCT images could also be used to measure axial length of the eye, and the corneal thickness, which may or not be thinner in the mutants.

Author’s Response: We sincerely appreciate the reviewer’s insightful and valuable suggestion, which we fully endorse as it helps enrich the characterization of the mutant ocular phenotype. As noted, our initial work focused on core ocular biometric parameters (equatorial diameter and lens diameter of enucleated eyes) and anterior chamber depth (measured via OCT in live mice). Regarding the potential measurement of axial length and corneal thickness from OCT images, we encountered a practical limitation: the optical coherence tomography (OCT) system at our experimental animal center is primarily configured for high-resolution imaging of local retinal structures. It lacks the specialized spectral-domain OCT module designed for small animals—characterized by sufficient axial resolution and scanning depth—required to accurately quantify axial length (a parameter demanding full ocular penetration). Consequently, we regret that axial length measurement was not feasible with our current equipment.

However, we fully adopted your suggestion to measure corneal thickness using the existing OCT images. In the revised manuscript, we have supplemented this analysis as follows: Corneal thickness was quantified from OCT images, and statistical comparisons between mutant and wild-type controls were performed. The results revealed no significant difference in corneal thickness between the mutant group and the control group (Fig 5B), indicating that the mutant eyes do not exhibit thinner corneas relative to wild-type littermates. We have made the following supplements in the revised manuscript Through measurement and statistical analysis of corneal thickness, the results showed that the mutant group did not exhibit significantly thinner corneas compared to the control group (Fig5 B). (Lines335-338).

The authors present evidence that there are no anatomical abnormalities in critical organs outside the eyes, but any description of the overall health of the mutants is lacking in the manuscript.

Author’s Response: We sincerely appreciate the reviewer’s astute observation, which has helped us improve the comprehensiveness of our study by supplementing critical information on the mutants’ systemic health. To address the gap of lacking descriptions on overall health status, we added a description of the overall health status of the mutant mice in the fifth section of the results section

To verify whether the mutant mice exhibit systemic health differences compared to wild-type controls, we conducted a comprehensive baseline health assessment. We found that that mutant mice and wild-type mice exhibit a well-groomed, sleek coat without lesions and normal exploratory behavior. Vital parameters, such as heart rate and body temperature, remain within established normal ranges. The animal exhibits a healthy appetite and consistent water consumption and stable body weight. Analysis of this baseline assessment demonstrated no statistically significant variation between the groups. (Lines353-360).

Visual function is assessed by the behavioral oculomotor test, which provides an insight into the overall function of the visual pathways, but does not provide any information on specific functions of the retinae of the mutants. An electroretinography (ERG) analysis would provide information on the overall function of the retina, the night vision and cone function of the retinas of the mutants. It is likely that there may be defects in both.

Author’s Response: We sincerely appreciate the reviewer’s insightful and constructive suggestion, which has significantly enhanced the depth of our retinal function characterization. We fully agree that electroretinography (ERG) is a critical tool to dissect specific retinal functions—including overall retinal integrity, night vision (rod-mediated function), and cone-mediated visual function—that cannot be captured by behavioral oculomotor tests alone. Following your guidance, we have supplemented comprehensive ERG analyses in the revised manuscript to address these key points: To further assess the specific visual function of the mutant retina, we performed ERG analysis under dark adaptation conditions using flash intensities of 1.5 cd·s/m² and 3.0 cd·s/m². The mutant mice exhibited abnormal ERG waveforms (Fig 6 A). The a-wave and b-wave amplitudes in the mutant groups were significantly reduced compared to the control group at flash intensities of 1.5 cd·s/m² and 3.0 cd·s/m², respectively (Fig 6 B). The results indicated that the mutant group exhibited significantly compromised visual function compared to the control group. However, the underlying mechanism for the observed visual impairment in mutant mice remains incompletely understood. This functional deficit may originate from intrinsic retinal pathology, or alternatively, result from the physical obstruction of light entry caused by pupillary atresia in mutant mice (Fig 6 C). (Lines339-349) Notably, as mentioned in the revised manuscript, we acknowledge a potential confounding factor: the presence of pupillary atresia in mutant mice (Fig 6C) may physically obstruct light entry into the retina, which could contribute to the observed ERG deficits alongside intrinsic retinal pathology.

In the Discussion section, we have expanded on this point to contextualize the findings: Also, to further characterize retinal function, we performed electroretinography (ERG), which revealed significant visual function impairment in the mutant mice. However, due to the presence of pupil atresia in some mutant mice, the specific cause of visual function impairment remains unclear. (Lines397-401) ERG deficits are consistent with the structural abnormalities identified via HE staining, but may also be partially attributed to pupillary atresia limiting light penetration. Future studies (e.g., after resolving the pupillary obstruction or via genetic validation of the causal gene) will help disentangle the relative contributions of these two factors to retinal functional impairment.

In Fig 5A there are micrographs of HE stained retinae of a mutant eye and control eye. By looking at these it appears that there are some differences in the thickness of some of the retinal layers, the photoreceptor and ONL seem thinner in the mutant, whilst the RPE appears thicker in the mutant. But this can only be solved by actually measuring the layers and make statistical comparisons. OCT images from the retinae of mutants and control could also be used for that purpose, i.e. to measure the thickness of layers, and if they are even across the retina or not. It appears from the micrographs that there is some variability there in the mutant.

Author’s Response: Thank you very much for your suggestions. We basically agree with your point of view. The retinal HE staining micrographs show that the thickness of certain layers of the retina in the mutant group is different from that in the control group. However, in some mutant mice, HE staining revealed severe disorganization of the outer nuclear layer accompanied by densely packed rosettes, precluding accurate thickness measurement; representative images are shown in Fig 7 A. Therefore, we quantified retinal rosettes in mutant mice to assess the severity of retinal pathology. We have made the following supplements in the revised manuscript Furthermore, the high density of retinal rosettes in certain regions of the mutant mouse retina makes it challenging to accurately measure the thickness of the outer nuclear layer (ONL). Therefore, we quantified the number of retinal rosettes at multiple time points. A pronounced retinal fold is observed near the optic disc, whereas the peripheral retina appears normal. The results indicated that the outer nuclear layer of the mutant group exhibited varying degrees of severe pathological alterations compared to the control group at various time points (Fig 7 C). Based on these results, we speculate that the pathological alterations in this mouse model were confined to the eyes and did not impact other regions

(Lines368-379). In addition, we have added retinal images of HE staining at other time points in Fig 7 A.

We appreciate you highlighting this issue. The figure has been readjusted and supplemented with other statistical results. Furthermore, we conducted additional measurements of RPE thickness and statistical analyses. We have made the following supplements in result: Histological analysis by HE staining demonstrated a significant thickening of the retinal pigment epithelium (RPE) in mutant mice compared with controls. (Fig 7 B). (Lines368-369) We have made the following supplements in discussion: The mutant mice exhibited a significantly thickened RPE layer compared with the control group. Given the multifaceted physiological roles of RPE cells, their dysfunction and degeneration are strongly implicated in the pathogenesis of multiple retinal degenerative disorders, including retinitis pigmentosa, Best disease, age-related macular degeneration, and Stargardt disease [35]. Best’s disease (BD), an autosomal dominant macular degeneration, is characterized by the accumulation of lipofuscin-like material within and beneath the retinal pigment epithelium (RPE), driving progressive central vision loss[36].In AMD, RPE dysfunction drives the progressive accumulation of basal deposits (drusen), leading to hypoxia, choroidal neovascularization, and eventual degeneration of the RPE and photoreceptor cells[37].In this study, the mutant mice also exhibited abnormalities in the RPE. The observed RPE thickening resembles the structural changes seen in related retinal diseases, further demonstrating that the mutant mice have defects in eye development which lead to impaired visual function. (Lines451-463)

Ocular developmental abnormalities in the mutant mice lead to pupillary atresia, which prevents mydriasis. The operational principle of optical coherence tomography (OCT) depends on adequate light penetration into the fundus following mydriasis, allowing for the visualization and quantitative measurement of the retinal layered structures. The inability to induce mydriasis in the mutant mice rendered effective light entry into the fundus unattainable, making OCT measurements unfeasible in this study; consequently, data on the thickness of individual retinal layers could not be obtained. This finding aligns with the ERG results, as the abnormal pupil morphology resulted in atypical ERG waveforms. Additionally, subsequent HE pathological findings confirmed that the mutant mice exhibit structural abnormalities in the retina. We have also included microscopic images taken before an

---

## [Decision Letter · Decision Letter 1]

17 Dec 2025

Characterization of a spontaneous microphthalmia-like mutant mouse with isolated ocular defects

PONE-D-25-46829R1

Dear Dr. Li,

We’re pleased to inform you that your manuscript has been judged scientifically suitable for publication and will be formally accepted for publication once it meets all outstanding technical requirements.

Kind regards,

Yevgenya Grinblat, Ph. D.

Academic Editor

PLOS One

Additional Editor Comments (optional):

Reviewers' comments:

Reviewer's Responses to Questions

**Comments to the Author**

Reviewer #1: All comments have been addressed

Reviewer #2: All comments have been addressed

2. Is the manuscript technically sound, and do the data support the conclusions?

Reviewer #1: Yes

Reviewer #2: Yes

3. Has the statistical analysis been performed appropriately and rigorously?

Reviewer #1: Yes

Reviewer #2: Yes

4. Have the authors made all data underlying the findings in their manuscript fully available?

Reviewer #1: Yes

Reviewer #2: Yes

5. Is the manuscript presented in an intelligible fashion and written in standard English?

Reviewer #1: Yes

Reviewer #2: Yes

Reviewer #1: The authors have fully addressed the issues I raised, and added data to the study that shed further light on the issues of concern.

Reviewer #2: Thank you for thoroughly addressing all of my concerns in this revision. Minor comment: please make sure to proofread the conclusions section for punctuation, and lines 488 - 489 (“we have restated it” seems out of place)

**Do you want your identity to be public for this peer review?** For information about this choice, including consent withdrawal, please see our Privacy Policy

Reviewer #1: No

Reviewer #2: No

---

## [Editor Report · Acceptance letter]

PONE-D-25-46829R1

PLOS One

Dear Dr. Li,

I'm pleased to inform you that your manuscript has been deemed suitable for publication in PLOS One. Congratulations! Your manuscript is now being handed over to our production team.

Kind regards,

on behalf of

Dr. Yevgenya Grinblat

Academic Editor

PLOS One